# Video Saliency Object Detection with Motion Quality Compensation

Hengsen Wang [1], Chenglizhao Chen [2], Linfeng Li [1] and Chong Peng [1,*]

1   School of Computer Science and Technology, Qingdao University, Qingdao 266071, China
2   School of Computer Science and Technology, China University of Petroleum, Qingdao 266580, China
*   Correspondence: pchong1991@163.com

**Abstract:** Video saliency object detection is one of the classic research problems in computer vision, yet existing works rarely focus on the impact of input quality on model performance. As optical flow is a key input for video saliency detection models, its quality significantly affects model performance. Traditional optical flow models only calculate the optical flow between two consecutive video frames, ignoring the motion state of objects over a period of time, leading to low-quality optical flow and reduced performance of video saliency object detection models. Therefore, this paper proposes a new optical flow model that improves the quality of optical flow by expanding the flow perception range and uses high-quality optical flow to enhance the performance of video saliency object detection models. Experimental results on the datasets show that the proposed optical flow model can significantly improve optical flow quality, with the S-M values on the DAVSOD dataset increasing by about 39%, 49%, and 44% compared to optical flow models such as PWCNet, SpyNet, and LFNet. In addition, experiments that fine-tuning the benchmark model LIMS demonstrate that improving input quality can further improve model performance.

**Keywords:** optical flow map; motion quality; long-term; video saliency object detection





## 1. Introduction

In recent years, Video Saliency Object Detection (VSOD) has received widespread research attention. Its main task is segmenting the most interesting target regions in videos and separating salient objects from their backgrounds in dynamic environments [1–5]. VSOD has been widely applied in many advanced computer vision tasks, such as video quality assessment [6], video compression [7], and face recognition [8].

Currently, image-based saliency object detection has been extensively studied. Unlike image-based saliency object detection that only uses spatial information within a single frame to predict saliency maps, VSOD needs to explore the motion information hidden in video sequences. However, many works [9–13] have not focused on the impact of motion information on saliency detection results. For example, Zhang et al. [14] proposed the determination of temporal and spatial misalignment by fusing the temporal alignment feature and spatial feature of adjacent frames. However, when the motion information is inaccurate, this method cannot solve the problem of temporal and spatial misalignment between adjacent frames. In [15], dense optical flow is merged to predict the position of vehicles on the road, but when the optical flow is uncertain, the prediction fails. Qi et al. [16] were able to effectively detect small objects through multi-scale modal fusion, but they overlooked the importance of motion information in the detection of small objects. Therefore, the focus of this paper is to improve the quality of optical flow and to address the negative impact of low-quality optical flow on the VSOD model.

Optical flow map is a computer vision technique used to track the motion and movement of objects within an image sequence or video. Here are some of the important applications and benefits of optical flow maps: object tracking [17,18], motion detection [19],

video compression [7], and other fields. Optical flow plays a crucial role in video saliency object detection models. It has several advantages: first, it reduces the focus on complex backgrounds and highlights salient regions of objects. Secondly, it allows for feature fusion and complements other features. Moreover, optical flow maps can be used for salient object localization and object mining [20]. Current mainstream methods for obtaining motion information include CRAFT [21], GMFlow [22], SpyNet [23], PWCNet [24], and LFNet [25]. Although these methods have excellent detection speed and accuracy, they may also generate optical flow maps with image generalization and contour blurring. This is because traditional optical flow calculation methods only rely on the image information of two adjacent frames to calculate the optical flow, which is a short-term strategy for computing optical flow, ignoring the motion states of objects in the past and future periods of time, as shown in Figure 1a. If an object does not move in a period of time, it will generate low-quality optical flow maps, as shown in the third column of Figure 1b; it is difficult for low-quality optical flow maps to provide effective position and motion information in video saliency object detection. The low-quality optical flow map hinders the performance improvement of video saliency object detection models. Therefore, to generate high-quality optical flow maps and obtain high-quality motion information, this paper proposes a new optical flow calculation model based on traditional optical flow models; the new optical flow model expands the range of optical flow perception, improving the quality of optical flow; the overall structure of the new optical flow model is shown in Figure 2. To address the limitations of traditional optical flow models, our main contributions are:

(1) Improve optical flow quality by expanding the perception range of optical flow. Instead of simply relying on adjacent frames to generate optical flow maps, we consider multiple frames.

(2) Select the optimal optical flow map through an optical flow perception module. This approach significantly enhances the quality of optical flow.

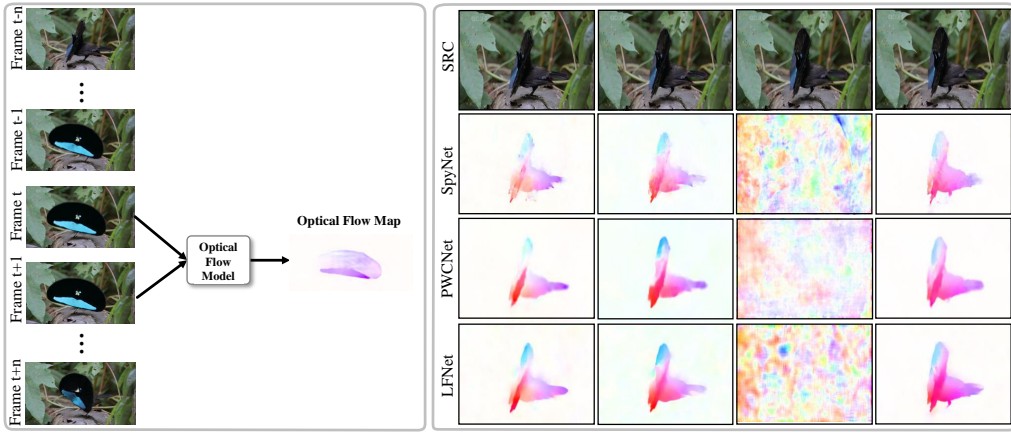

(a) Traditional optical flow model      (b) Optical flow maps generated by traditional optical flow models

**Figure 1.** (**a**) It represents traditional optical flow models that only use two frames of images for optical flow calculation. (**b**) Optical flow maps generated by the traditional optical flow models.

As a key input of video saliency detection models, the quality of optical flow has rarely been studied in its impact on model performance. To address the limitations of traditional optical flow models that only generate flow maps using adjacent frames, this paper proposes to improve the quality of optical flow by expanding its perception range. To demonstrate that the new optical flow calculation method can improve flow quality and that improved flow quality can effectively enhance the performance of video saliency object detection models, a motion quality compensation-based video saliency object detection method is proposed. As shown in Figure 2, this method mainly consists of two stages; in the first stage, the quality of optical flow is improved using the new optical flow

model; in the second stage, an existing video saliency object detection model, LIMS [26], is fine-tuned based on the high-quality optical flow maps to obtain the final video saliency object detection results. The experimental results demonstrate that the new optical flow calculation method proposed in this paper can effectively improve the quality of optical flow, and high-quality optical flow maps can improve the performance of video saliency object detection models.

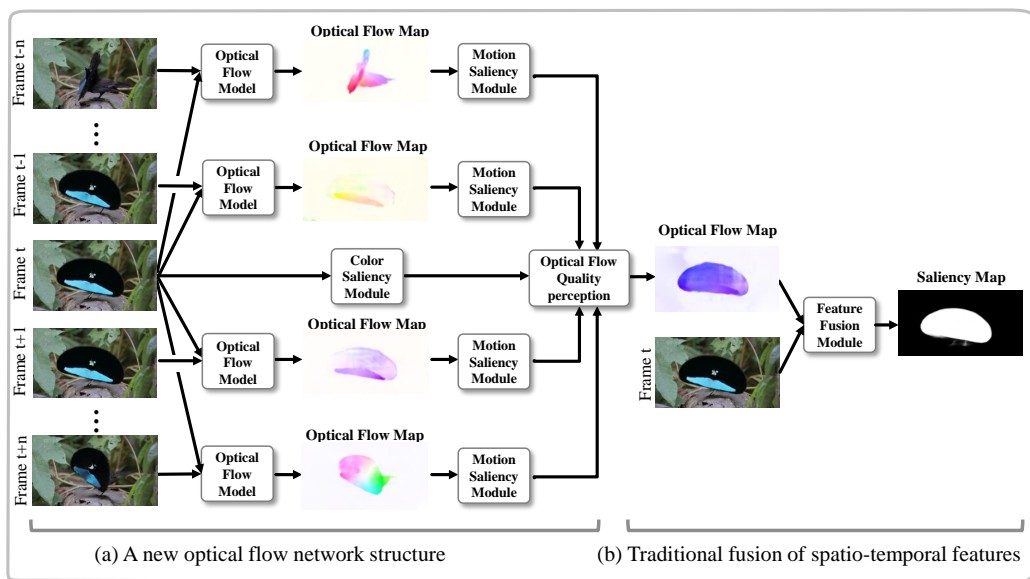

**Figure 2.** The (**a**) The new optical flow model proposed in this paper. (**b**) Fused high-quality optical flow maps and RGB images using a traditional feature fusion module to obtain a saliency map.

## 2. Related Work

### 2.1. Traditional Optical Flow Model

Optical flow computation is a classic research problem in computer vision, and traditional optical flow algorithms include LkFlow [27], PCAFlow [28], EpicFlow [29], and so on. Generally, traditional optical flow algorithms assume that a pixel and its surrounding pixels will not experience sudden changes in brightness or position over time. However, this assumption limits the accuracy and computational speed of traditional optical flow algorithms, which, in turn, restricts their practical applications in real-life situations.

### 2.2. Deep Learning-Based Optical Flow Model

The rise of deep learning has brought breakthrough development to the optical flow field, and optical flow models based on deep learning surpass traditional models in terms of speed and accuracy. Based on deep learning, optical flow models are mainly divided into supervised learning and unsupervised learning. In terms of supervised learning for optical flow models, Dosovitskiy et al. [30] first introduced Convolutional Neural Networks (CNN) to the optical flow field and proposed the FlowNet network structure. It inputs two frames of images and uses CNN for end-to-end training to output an optical flow map of the original image size. The accuracy of FlowNet is slightly lower than that of traditional optical flow models, but the speed is much faster. Xu et al. [22] proposed a GMF framework that has a higher resolution for residual flow detection. Ranjan et al. [23] proposed the SpyNet optical flow model, which uses the pyramid concept in traditional saliency object detection models to design the optical flow model from coarse to fine. Sun et al. [24] proposed the PWCNet optical flow model, which adds the cost volume concept in classical methods based on the pyramid concept to improve network performance and become the benchmark for new optical flow models. Hui et al. [25] proposed the LFNet optical flow model, which introduces the idea of feature regularization based on the cost volume to further improve the quality of optical flow maps. In terms of unsupervised

optical flow models, Liu et al. [31] proposed the DDFlow optical flow model, which generates reliable predictions through a teacher network and guides the student network to learn optical flow. Ding et al. [32] proposed the EFC optical flow model, which jointly learns video segmentation and optical flow estimation, and the two mutually guide and improve the performance of video segmentation and optical flow models.

### 2.3. Video Saliency Object Detection Using Optical Flow

Currently, video saliency object detection models mainly fuse spatial and temporal features to obtain saliency maps or use motion and position information provided by optical flow to locate salient objects and generate saliency maps. Li et al. [33] proposed using optical flow as network input to calculate motion saliency. Through a motion attention module, unreliable motion clues are filtered out before spatial and temporal feature fusion. Zhou et al. [34] proposed a novel end-to-end camera-agnostic video segmentation network, which is based on a traditional two-stream network but with a new module designed to enable interaction between temporal and spatial information. This method uses motion information as a rough indicator to locate potential target objects. Ren et al. [35] proposed a novel three-stream network that includes a spatio-temporal network, a spatial network, and a temporal network, the saliency clues calculated from the spatio-temporal network are used as the attention mechanism to enhance the main network. In order to improve the accuracy of video saliency detection, Chen et al. [36] proposed a new spatio-temporal saliency consistency modeling and learning method. In the paper [26], salient object proposals and key frames are selected based on motion clues provided by optical flow. Optical flow plays an important role in the video saliency object detection model as a critical input to the network, and the quality of optical flow has a crucial impact on the model's performance.

### 3. Method Overview

#### 3.1. Existing Optical Flow Models

Currently, the main way to obtain motion information is through optical flow. The principle of existing optical flow calculation tools is shown in Figure 1a. The tools take in adjacent frames of a video segment $\mathbf{F} = \{\mathbf{I}_1, \mathbf{I}_2, ..., \mathbf{I}_u\}$ with $u$ frames and output an optical flow map. The principle is to establish a relationship between the pixels between adjacent frames $\mathbf{I}_t$ and $\mathbf{I}_{t+1}$ in the video segment and calculate the optical flow based on this correlation, as shown in Equation (1):

$$\mathbf{of}_i = \mathrm{vi}\Big\{ \mathrm{FLOWNet}(\mathbf{I}_t, \mathbf{I}_{t+1}) \Big\}, \tag{1}$$

FLOWNet(.) represents existing optical flow calculation tools, such as SpyNet [23], PWCNet [24], and LFNet [25], which rely on two consecutive video frames as input to calculate the spatial displacement of the same pixel region in the vertical and horizontal directions. The optical flow map of the t-th frame is obtained through commonly used visualization method vi{.}. Using Equation (1), the optical flow map of each pair of frames in $\mathbf{F} = \{\mathbf{I}_1, \mathbf{I}_2, ..., \mathbf{I}_u\}$ is sequentially calculated to obtain the optical flow $\mathbf{OF} = \{\mathbf{of}_1, \mathbf{of}_2, ..., \mathbf{of}_{u-1}\}$, as shown in Figure 1b.

Traditional optical flow calculation methods assume that objects in the video move slowly between adjacent frames. If a salient object in the video remains still for a short period of time, it is difficult to obtain the displacement information of the object in the vertical and horizontal directions using only two adjacent frames to calculate the optical flow, resulting in a low-quality optical flow map. As shown in the third column of Figure 1b.

#### 3.2. A Novel Optical Flow Model

Existing optical flow models rely only on generating optical flow between adjacent frames in a video segment, which is a short-term method for calculating optical flow. To address the inherent limitations of short-term optical flow calculation methods, inspired

by the pedestrian trajectory calculation method in paper [37], this paper proposes a new optical flow calculation method. The new optical flow model is structured as shown in Figure 2a and consists of four parts: (1) color saliency module, (2) motion saliency module, (3) computational approach for expanding the optical flow perception range, (4) and optical flow quality perception module.

### 3.2.1. Motion Saliency Module

The proposed motion saliency module in this paper aims to better assess the quality of optical flow maps by converting them into motion saliency maps. The unified storage format of color saliency maps and motion saliency maps facilitates various computational operations on subsequent optical flow maps.

The motion saliency module uses the optical flow $\mathbf{OF} = \{\mathbf{of}_1, \mathbf{of}_2, ..., \mathbf{of}_{u-1}\}$ and its corresponding ground truth map $\mathbf{GT} = \{\mathbf{gt}_1, \mathbf{gt}_2, ..., \mathbf{gt}_{u-1}\}$ from the MSRA10K [38] dataset to fine-tune an image saliency detection model (ISOD) to generate the motion saliency map. By using a loss function to correct the difference between predicted values and ground truth, the model can generate more accurate motion saliency maps. The loss function (**MS**) for fine-tuning the ISOD model is shown in Equation (2), with a learning rate of $10^{-7}$,

$$L_{\mathbf{MS}} = -\sum_{t=1}^{u-1} \mathbf{gt}_t \times \log \mathbf{of}_t + (1 - \mathbf{gt}_t) \times \log(1 - \mathbf{of}_t), \tag{2}$$

after fine-tuning, the optical flow maps $\mathbf{OF} = \{\mathbf{of}_1, \mathbf{of}_2, ..., \mathbf{of}_{u-1}\}$ are fed into the fine-tuned image saliency detection model to generate motion saliency maps $\mathbf{MS} = \{\mathbf{ms}_1, \mathbf{ms}_2, ..., \mathbf{ms}_{u-1}\}$. The general process can be represented by Equation (3):

$$\mathbf{MS} = M(\Theta, \mathbf{OF}), \tag{3}$$

M(.) represents the fine-tuned ISOD model, which in this paper, is CPD [39]; $\Theta$ is the model parameter obtained by training the model through Equation (2) on the MSRA10k dataset. The optical flow maps and generated motion saliency maps of the motion saliency module are shown in rows three and four of Figure 3.

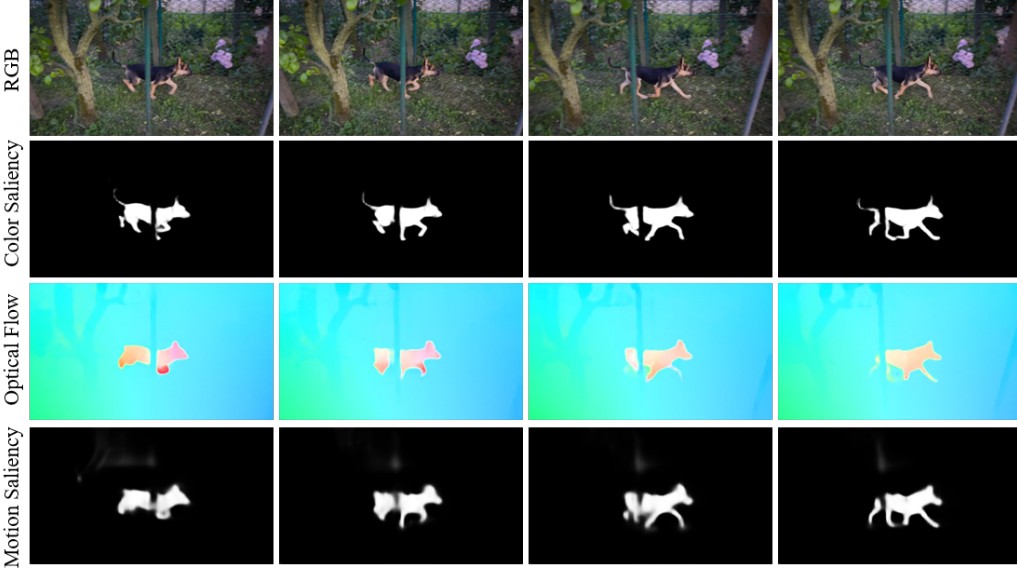

**Figure 3.** Color saliency is obtained from RGB images through the color saliency module, while motion saliency maps are obtained from optical flow maps through the motion saliency module.

### 3.2.2. Color Saliency Module

One of the main functions of the color saliency module is to convert RGB images into color saliency maps that contain clear saliency structures, which prepares them for subsequent optical flow quality evaluation. We use the RGB images $\mathbf{F} = \{\mathbf{I}_1, \mathbf{I}_2, ..., \mathbf{I}_u\}$ and their corresponding ground truth maps $\mathbf{GT} = \{\mathbf{gt}_1, \mathbf{gt}_2, ..., \mathbf{gt}_u\}$ in the MSRA10K [38] dataset. By using the loss function (**CS**) to correct the difference between RGB and ground truth, the model can generate more accurate color saliency maps; the loss function (**CS**) for fine-tuning the ISOD model is shown in Equation (4),

$$\mathrm{L_{cs}} = -\sum_{t=1}^{u} \mathbf{gt}_t \times \log\mathbf{I}_t + (1 - \mathbf{gt}_t) \times \log(1 - \mathbf{I}_t), \tag{4}$$

inputting $\mathbf{F} = \{\mathbf{I}_1, \mathbf{I}_2, ..., \mathbf{I}_u\}$ yields the color saliency map $\mathbf{CS} = \{\mathbf{cs}_1, \mathbf{cs}_2, ..., \mathbf{cs}_u\}$, as shown in Equation (5),

$$\mathbf{CS} = \mathrm{M}(\Omega, \mathbf{F}), \tag{5}$$

$\Omega$ is the model parameter obtained by training the model through Equation (4), $\mathrm{M}(.)$ represents the fine-tuned ISOD model, which in this paper, is CPD [39]. As shown in the first and second rows of Figure 3, the input RGB image and output color saliency map of the color saliency module is displayed, respectively.

### 3.2.3. Enlarging the Optical Flow Perception Range

The main contribution of the proposed optical flow model in this article is to solve the problem of traditional optical flow calculation methods being unable to obtain motion information of stationary objects by expanding the perception range of optical flow. By expanding the perception range of optical flow, the proposed optical model in this paper can perceive the position changes of stationary objects in other frames, making it easier to transform them into moving objects and calculate their motion state.

The method of expanding the perception range of optical flow is a long-term approach that can solve the problem of traditional optical flow calculation methods being unable to obtain the motion state of salient stationary objects in a short period of time. The basic idea of this method is to expand the visual field of optical flow calculation to multiple frames and obtain the motion state of objects by comparing the pixel value changes between multiple frames. This can avoid the problems encountered by existing optical flow models and improve the stability and accuracy of optical flow calculation.

This paper proposes a new optical flow model to obtain high-quality optical flow by expanding the optical flow perception range for each frame in a *u* frame video segment. As shown in the left half of Figure 2, the method involves calculating 2*n* optical flow frames for Frame t by comparing it with the previous *n* frames and the subsequent *n* frames and then selecting the best optical flow frame using an optical flow quality perception module. Although this approach can yield high-quality optical flow, it requires a significant amount of memory and time to compute 2*n* optical flow frames for each frame. To address this issue, two steps are proposed: (1) to determine the optical flow frames that require expanding the perception range based on optical flow quality module; and (2) to determine the range of optical flow perception (i.e., *n* value).

(1) Video segment $\mathbf{F}$ obtained optical flow $\mathbf{OF} = \{\mathbf{of}_1, \mathbf{of}_2, ..., \mathbf{of}_{u-1}\}$ using existing optical flow models and determined the optical flow maps that need to be expanded for the optical flow perception range by the optical flow quality perception module, as shown in Equations (7) and (8). For the optical flow map, $\mathbf{of}_t$, it expands its optical flow perception range to obtain 2*n* optical flow maps, as shown in Equation (6):

$$\left\{ \mathrm{UF}(\mathbf{I}_t, \mathbf{I}_{t-n}), ..., \mathrm{UF}(\mathbf{I}_t, \mathbf{I}_{t+n}) \right\} = \left\{ \mathbf{uf}_{t-n}, ..., \mathbf{uf}_t, ..., \mathbf{uf}_{t+n} \right\}, \tag{6}$$

$\mathrm{UF}(.)$ stands for the method proposed in this paper for expanding the optical flow perception range. By taking the *t*-th frame as the center, feeding the previous *n* frames

and the following *n* frames into the optical flow model, and computing the optical flow with the *t*-th frame, 2*n* optical flow maps can be obtained; it calculates a total of 2*n* frames of optical flow images for both $\mathbf{I}_t$ and $\mathbf{I}_{t-n}, ..., \mathbf{I}_t, ..., \mathbf{I}_{t+n}$. Through  Equation (9) in the optical flow quality perception module, the best optical flow map among the 2*n* frames is selected to replace the original optical flow image $\mathbf{of}_t$. The visualized results of the replaced optical flow are shown in the fifth row in  Figure 4. Compared with other traditional optical flow models, the UF(.) method greatly improves the quality of optical flow. For example, in Figure 4, it is difficult for traditional optical flow models to obtain high-quality optical flow for the bird in the first column when it is still; the person riding a bicycle in the second column is in a stationary state when reaching the highest point, and traditional optical flow models cannot produce high-quality optical flow; the traditional optical flow model also fails to generate high-quality optical flow when the frog is stationary before jumping. The optical flow model proposed in this paper effectively improves the quality of optical flow in the aforementioned scenarios.  By determining the optical flow map that needs to expand the perception range, this targeted expansion strategy greatly reduces the time and memory consumption of the optical flow calculation method, although expanding the optical flow perception range can effectively improve the quality of optical flow. There can also be some failure cases, such as in the birdfall video sequence of the Segtrack-v2 dataset; due to the small size and fast motion of the salient object in this video sequence, as well as the low pixel resolution, it is easy to generate low-quality optical flow maps, and methods to expand the optical flow perception range cannot effectively solve the problem. In the Libby video sequence of the Davis dataset, the salient object is occluded by the occluding object, and it is impossible to generate high-quality optical flow maps and saliency maps. The occlusion problem is a challenge that most works need to address.

(2) Determining the size of the range to expand optical flow perception. Multiple sets of experiments have shown that the optical flow perception range is not necessarily better when it is larger. That is, when the value of *n* is too large, the quality improvement of optical flow is limited. When $n = 4$, the quality improvement of optical flow is relatively large with small time and space costs. Specific comparative results are presented in Section 5.1.

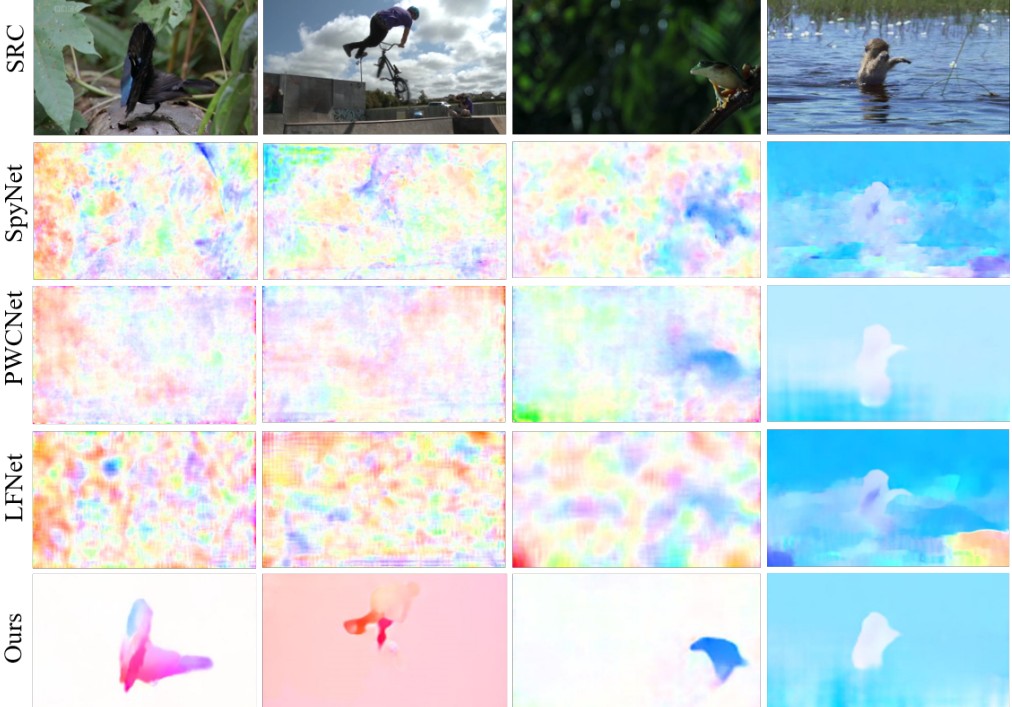

**Figure 4.** Comparison of optical flow maps generated by different optical flows in multiple scenes, "Ours" represents the optical flow map generated by the optical flow model proposed in this paper.

### 3.2.4. Optical Flow Quality Perception Module

Traditional optical flow models are prone to generate low-quality optical flow images. The main function of the optical flow quality perception module is: (1) to filter out the low-quality optical flow maps generated by traditional optical flow calculation methods; (2) to select high-quality optical flow maps.

(1) For video segment $\mathbf{F}$, the color saliency module is used to generate the color saliency map $\mathbf{CS} = \{\mathbf{cs}_1, \mathbf{cs}_2, ..., \mathbf{cs}_u\}$. The optical flow map $\mathbf{OF} = \{\mathbf{of}_1, \mathbf{of}_2, ..., \mathbf{of}_{u-1}\}$ generated by the traditional optical flow model is used to obtain the corresponding motion saliency map $\mathbf{MS} = \{\mathbf{ms}_1, \mathbf{ms}_2, ..., \mathbf{ms}_{u-1}\}$ through the motion saliency module. Calculating the average S-M [40] score of $\mathbf{MS}$ and $\mathbf{CS}$ through Equation (7),

$$\mathrm{avgSM} = \frac{1}{u-1} \sum_{t=1}^{u-1} \mathrm{SM}(\mathbf{ms}_t, \mathbf{cs}_t), \tag{7}$$

SM(.) is used to calculate the S-M score of the corresponding motion saliency map $\mathbf{ms}_t$ and color saliency map $\mathbf{cs}_t$. avgSM is the average of the S-M score of $\mathbf{MS}$ and $\mathbf{CS}$.

This paper chooses S-M [40] to perceive the quality of optical flow, which is different from other quality metrics such as maxF [41] and MAE [42] that are based on pixel-level errors and ignore the importance of structural similarity. As shown in Figure 5, high-quality motion saliency maps generally have higher structural consistency with color saliency maps. Therefore, an S-M that emphasizes structural similarity is used to evaluate the quality of optical flow in this paper. Generally, a high-quality motion saliency map and color saliency map have high S-M scores. Conversely, if the S-M scores of a motion saliency map and its corresponding color saliency map are low, then the quality of the motion saliency map is considered to be low. The quality of the motion saliency map is proportional to that of the optical flow. avgSM as a quality threshold if the S-M scores of a motion saliency map and its corresponding color saliency map in a certain frame satisfy Equation (8),

$$\mathrm{avgSM} \geq \mathrm{SM}(\mathbf{ms}_t, \mathbf{cs}_t), \tag{8}$$

then the optical flow corresponding to the motion saliency map is considered to be of low quality and the range of optical flow perception is expanded to optimize it.

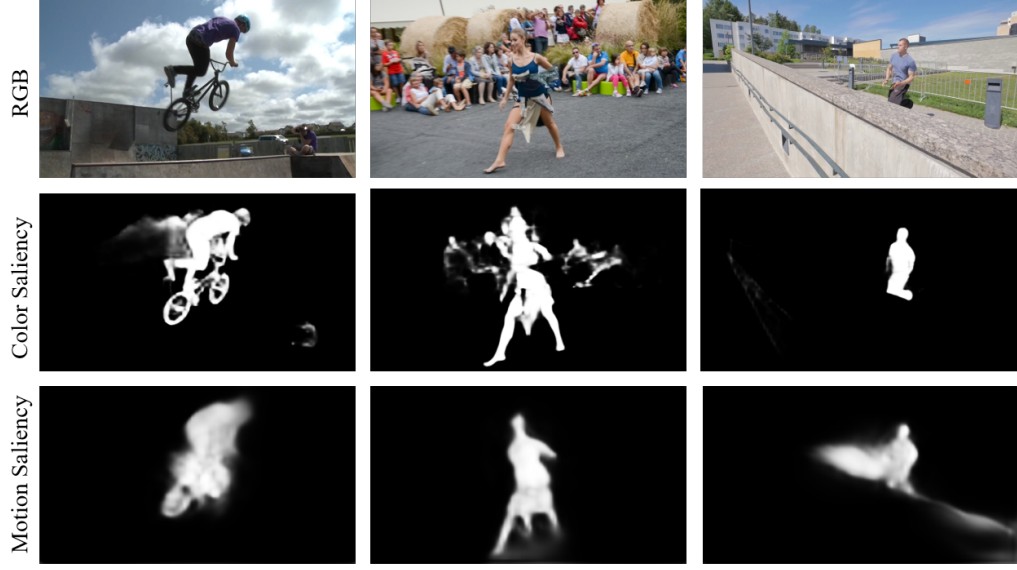

**Figure 5.** High-quality motion saliency maps and color saliency maps often have higher consistency in structure.

(2) The 2*n* optical flow maps obtained from Equation (6) are filtered using Equations (7) and (8) to select the optimal optical flow map to replace the original optical flow map $\mathbf{of}_t$, as shown in Equation (9):

$$\mathbf{of}_t = \max\left\{ \text{SM}\left(\text{M}(\mathbf{uf}_{t-n}), \mathbf{cs}_t\right), ..., \text{SM}\left(\text{M}(\mathbf{uf}_t), \mathbf{cs}_t\right), ...\text{SM}\left(\text{M}(\mathbf{uf}_{t+n}), \mathbf{cs}_t\right) \right\}, \quad (9)$$

M(.) is the method for generating motion saliency maps represented by Equation (2) and Equation (3); SM(.) is used to calculate the S-M score of the motion saliency map and the color saliency map to measure the quality; max{.} represents selecting the optical flow map contained in the highest SM(.) score as the optimal optical flow map to replace $\mathbf{of}_t$.

In this paper, a comparison of motion saliency maps obtained from the proposed optical flow model and other optical flow models is shown in Figure 6. The proposed optical flow model is capable of improving the quality of motion saliency maps specifically, and the performance of the motion saliency maps is significantly improved compared to existing optical flow models such as SpyNet [23], PWCNet [24], and LFNet [25] on five benchmark datasets, as shown in Section 5.2.1. Then, the existing LIMS [26] model is fine-tuned using motion saliency maps obtained from SpyNet, PWCNet, LFNet, and the proposed optical flow model, and the impact of different qualities of motion saliency maps on the performance of the LIMS model is compared, as presented in Section 5.2.1.

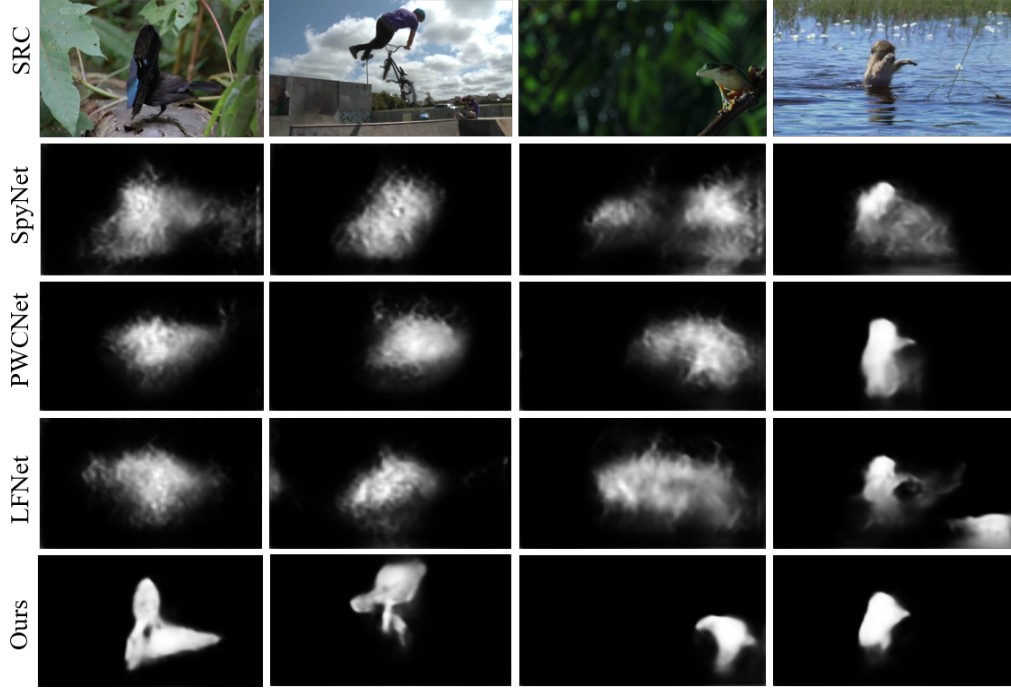

**Figure 6.** Comparison of motion saliency maps generated by four different optical flow models, with "Ours" representing the motion saliency map generated by the optical flow model proposed in this paper.

## 4. Experiments

### 4.1. Datasets

Based on the traditional experimental settings, this paper evaluates the proposed optical flow model on five widely used public datasets, including Davis [43], Segtrack-v2 [44], Visal [45], DAVSOD [46], and VOS [47].

The Davis dataset contains 50 video sequences totaling 3455 frames, with most sequences containing moderate motion and each video providing accurate manually labeled tags.

The Segtrack-v2 dataset contains 13 video sequences totaling 1024 frames, with complex backgrounds and changing motion patterns, often more challenging than the Davis dataset.

The Visal dataset contains 17 video sequences totaling 963 frames, with pixel-level annotated labels given every 5 frames. This dataset is relatively simple, focusing mainly on spatial information and rarely having intense motion.

The DAVSOD dataset contains 226 video sequences totaling 23,938 frames, which is the most challenging dataset, involving various object instances, different motion patterns, and saliency transfers between different objects.

The VOS dataset contains 40 video sequences totaling 24,177 frames, with only 1540 frames well annotated, and most of the video sequences are obtained in indoor scenes.

### 4.2. Experimental Environment

This paper's experiments were conducted using the Pytorch deep learning framework and Python programming language on a GTX1080Ti workstation. The motion saliency module was fine-tuned on CPD [39] using the optical flow maps and corresponding ground truth maps from MSRA10K [38]. PWCNet [24] was selected as the optical flow calculation tool. The LIMS [26] model was chosen for fine-tuning.

### 4.3. Evaluation Metrics

To accurately measure the consistency between the saliency maps generated by video salient object detection (VSOD) and the manually annotated ground truth maps, this paper adopts three commonly used evaluation metrics in video saliency object detection, including maxF [41], MAE [42], and S-M [40].

Precision (P) and Recall (R) are commonly used evaluation metrics, but sometimes they can be contradictory. When both values are low, they may not effectively represent the performance of the results. MaxF takes both P and R into account, and its formula is shown in Equation (10):

$$\text{maxF}_\beta = \frac{(1+\beta^2) \times P \times R}{\beta^2 \times P + R}, \tag{10}$$

In this paper, $\beta^2$ is set to 0.3 to calculate the MaxF value between the predicted and ground truth maps. When the MaxF value is high, it indicates that the predicted map is closer to the ground truth map.

Mean Absolute Error (MAE) represents the average of absolute errors between predicted and true values, as shown in Equation (11):

$$\text{MAE} = \frac{1}{W \times H} \sum_{x=1}^{W} \sum_{y=1}^{H} |S(x,y) - G(x,y)|, \tag{11}$$

where W and H represent the width and height of the predicted map and ground truth map, respectively; S(x,y) denotes the predicted saliency value at pixel (x,y), and G(x,y) denotes the value at pixel (x,y) in the ground truth map.

Structure Measure (S-M) is defined to evaluate the structural similarity of foreground maps, which is used to assess errors based on regions and objects, as shown in Equation (12):

$$\text{S-M} = \alpha \times S_m + (1 - \alpha) \times S_r, \tag{12}$$

where $S_r$ is the structural similarity measure for region-based errors, $S_m$ is the structural similarity measure for object-based errors, and $\alpha$ is usually set to 0.5.

## 5. Experimental Results Analysis

### 5.1. Ablation Experiment for the Parameter n

As an innovation in this paper, the expansion of the optical flow perception range is represented by the parameter $n$. Generally, the larger the value of $n$, the greater the improvement in optical flow quality; the smaller the value of $n$, the lower the time-space cost.

As shown in Table 1, the situation is not always like this, and there may be two reasons. First, in the Davis and Visual datasets, there are clear boundaries between salient objects and the background, and the optimal results can be obtained within a smaller perception range. Conversely, in the Segtrack-v2 dataset, the salient objects are small and have different motion patterns, requiring a larger perception range to compute the best optical flow map. Secondly, selecting the best optical flow map through the quality perception module is not always reliable. Due to these two factors, the optimal value of $n$ is uncertain. Taking into account time, space cost, and accuracy, this paper chooses $n$ to be 4.

**Table 1.** Quality of optical flow maps with different perception ranges, the optimal data are represented in bold.

| Dataset | Metrics | $n = 2$ | $n = 4$ | $n = 6$ | $n = 8$ |
|---|---|---|---|---|---|
| Davis [43] | maxF | 0.787 | **0.798** | 0.790 | 0.789 |
| | SM | 0.844 | **0.855** | 0.848 | 0.846 |
| | MAE | 0.049 | **0.044** | 0.048 | 0.047 |
| Segtrack-v2 [44] | maxF | 0.648 | 0.699 | **0.701** | 0.695 |
| | SM | 0.760 | 0.791 | **0.795** | 0.790 |
| | MAE | 0.054 | 0.045 | **0.043** | 0.047 |
| Visal [45] | maxF | 0.624 | **0.734** | 0.722 | 0.725 |
| | SM | 0.736 | **0.796** | 0.786 | 0.790 |
| | MAE | 0.079 | **0.066** | 0.070 | 0.069 |

### 5.2. Validity Analysis of the Proposed Optical Flow Model

5.2.1. Quantitative Analysis

(1) Performance comparison of different optical flow models. Optical flow is susceptible to noise and lighting variations, which can lead to a decrease in the accuracy and stability of optical flow estimation. In this paper, we performed brightness equalization on the video sequences from five datasets before calculating optical flow to ensure that the images are more consistent in brightness. Additionally, this section is to compare the quality of optical flow between traditional optical flow models and the optical flow model proposed in this paper. To better demonstrate that expanding the optical flow perception range can improve the quality of optical flow, this paper only needs to ensure that the quality of the input video sequences for the optical flow models is the same. Therefore, in order to quantitatively analyze the effectiveness of the proposed new optical flow method in improving the quality of the motion saliency map, compared with CRAFT [21], GMFlow [22], SpyNet [23], PWCNet [24], and LFNet [25], optical flow models on five benchmark datasets, as shown in Table 2. Bold indicates the best data. According to Table 2, the proposed method can effectively improve the quality of the motion saliency map, especially for DAVSOD and VOS datasets.

**Table 2.** Performance comparison of different optical flow models, bold indicates optimal data. "Ours" represents the optical flow model proposed in this paper.

| Dataset | Davis [43] | | | Segtrack-v2 [44] | | | Visal [45] | | | DAVSOD [46] | | | VOS [47] | | |
|---|---|---|---|---|---|---|---|---|---|---|---|---|---|---|---|
| Metrics | maxF | S-M | MAE | maxF | S-M | MAE | maxF | S-M | MAE | maxF | S-M | MAE | maxF | S-M | MAE |
| Ours | **0.798** | **0.855** | **0.044** | **0.699** | **0.791** | **0.045** | **0.734** | **0.796** | **0.066** | **0.798** | **0.855** | **0.044** | **0.699** | **0.791** | **0.045** |
| CRAFT [21] | 0.795 | 0.850 | 0.044 | 0.695 | 0.789 | 0.048 | 0.731 | 0.793 | 0.069 | 0.793 | 0.848 | 0.046 | 0.695 | 0.688 | 0.048 |
| GMFlow [22] | 0.792 | 0.847 | 0.046 | 0.690 | 0.787 | 0.050 | 0.730 | 0.792 | 0.071 | 0.790 | 0.842 | 0.048 | 0.691 | 0.685 | 0.049 |
| PWCNet [24] | 0.787 | 0.844 | 0.049 | 0.648 | 0.760 | 0.054 | 0.624 | 0.736 | 0.079 | 0.450 | 0.613 | 0.148 | 0.405 | 0.566 | 0.167 |
| SpyNet [23] | 0.727 | 0.801 | 0.065 | 0.596 | 0.733 | 0.078 | 0.659 | 0.762 | 0.092 | 0.382 | 0.574 | 0.182 | 0.403 | 0.562 | 0.188 |
| LFNet [25] | 0.781 | 0.843 | 0.049 | 0.656 | 0.766 | 0.059 | 0.674 | 0.764 | 0.081 | 0.408 | 0.592 | 0.168 | 0.380 | 0.551 | 0.189 |

(2) Comparison of time consumption among different optical flow models. The comparison of time consumption between the proposed optical flow model and traditional opti-

cal flow methods on five benchmark datasets is shown in Table 3, with the time unit in minutes. Compared with existing optical flow models, the proposed model employs a strategy of expanding the optical flow perception range, which effectively improves the quality of optical flow with a limited increase in time consumption. Especially for the VOS dataset, the time consumption of the proposed model increased by 29% compared to the benchmark model PWCNet [24], but the key performance metrics S-M [40] of optical flow quality increased from 0.566 to 0.791, with an improvement of 40% in the S-M performance metric. As shown in Table 4, the method proposed in this paper improves the quality of optical flow compared to existing methods but also increases memory usage.

**Table 3.** The total time taken by different optical flow models to generate optical flow for different datasets, with the time unit in minutes, "Ours" represents the optical flow model proposed in this paper.

| Dataset | Davis [43] | B | Visal [45] | DAVSOD [46] | VOS [47] |
|---|---|---|---|---|---|
| Ours | 11 | 9 | 8 | 20 | 35 |
| CRAFT [21] | 9 | 7 | 6 | 15 | 26 |
| GMFlow [22] | 9 | 8 | 7 | 16 | 28 |
| SpyNet [23] | 8 | 6 | 5 | 16 | 28 |
| PWCNet [24] | 9 | 7 | 6 | 15 | 27 |
| LFNet [25] | 10 | 8 | 7 | 17 | 28 |

**Table 4.** The total memory space usage taken by different optical flow models to generate optical flow for different datasets, with the memory space unit in megabytes, "Ours" represents the optical flow model proposed in this paper.

| Dataset | Davis [43] | Segtrack-v2 [44] | Visal [45] | DAVSOD [46] | VOS [47] |
|---|---|---|---|---|---|
| Ours | 24 | 12 | 11 | 89 | 476 |
| CRAFT [21] | 20 | 9 | 8 | 78 | 356 |
| GMFlow [22] | 21 | 9 | 8 | 77 | 351 |
| SpyNet [23] | 21 | 10 | 8 | 79 | 360 |
| PWCNet [24] | 18 | 9 | 6 | 54 | 350 |
| LFNet [25] | 19 | 10 | 9 | 61 | 345 |

(3) Comparison of performance for fine-tuning the LIMS [26] model with different optical flow models. To demonstrate that the motion saliency maps generated by the proposed new optical flow model can improve the performance of video saliency object detection models, selecting LIMS [26] as the benchmark model for video saliency detection, this paper uses the motion saliency maps generated by the proposed optical flow model, CRAFT [21], GMFlow [22], SpyNet [23], PWCNet [24], and LFNet [25] to fine-tune the LIMS model and generate the final saliency map. The performance metrics of fine-tuned LIMS model using different optical flow models are compared in Table 5, where "+" represents fine-tuning the LIMS model using different optical flow models, and "Ours" represents our optical flow model, with the best indicators in bold. The performance metrics of the proposed fine-tuned optical flow LIMS model are superior to other optical flow models on the five benchmark datasets, with a significant improvement compared to the fine-tuning results of other optical flow models on the VOS dataset.

(4) The comparison of running speed (in frames per second) of the LIMS [26] model fine-tuned with motion saliency maps generated by different optical flow models is shown in Table 6. "+" represents fine-tuning the LIMS model using different optical flow models, and "Ours" refers to the optical flow model proposed in this paper. The baseline running speed of LIMS is 23 frames/s, and the running speed of the LIMS model fine-tuned by our proposed method is 24 frames/s. Compared with the running speed of other optical flow algorithms fine-tuned with the LIMS model, the running speed is improved. Although the optical flow model proposed in this paper increases the algorithm complexity, compared with other optical flow models fine-tuned with the LIMS model, the quality improvement

of motion information enhances the performance and running speed of the video saliency object detection model.

**Table 5.** Comparison of performance for fine-tuning LIMS model with different optical flow models, "+" represents fine-tuning the LIMS model using different optical flow models, and "Ours" represents our optical flow model, with the best metrics shown in bold.

| Dataset | Davis [43] | | | Segtrack-v2 [44] | | | Visal [45] | | | DAVSOD [46] | | | VOS [47] | | |
|---|---|---|---|---|---|---|---|---|---|---|---|---|---|---|---|
| Metrics | maxF | S-M | MAE | maxF | S-M | MAE | maxF | S-M | MAE | maxF | S-M | MAE | maxF | S-M | MAE |
| +Ours | **0.914** | **0.924** | **0.014** | **0.881** | **0.910** | **0.013** | **0.955** | **0.948** | **0.011** | **0.735** | **0.799** | **0.060** | **0.832** | **0.852** | **0.058** |
| +CRAFT [21] | 0.913 | 0.923 | 0.015 | 0.877 | 0.907 | 0.015 | 0.952 | 0.946 | 0.013 | 0.727 | 0.795 | 0.063 | 0.811 | 0.845 | 0.063 |
| +GMFlow [22] | 0.912 | 0.922 | 0.016 | 0.879 | 0.908 | 0.014 | 0.953 | 0.947 | 0.012 | 0.729 | 0.797 | 0.063 | 0.815 | 0.849 | 0.060 |
| +PWCNet [24] | 0.913 | **0.924** | 0.015 | 0.875 | 0.905 | 0.015 | 0.950 | 0.945 | 0.012 | 0.723 | 0.794 | 0.064 | 0.801 | 0.841 | 0.067 |
| +SpyNet [23] | 0.910 | 0.922 | 0.015 | 0.861 | 0.899 | 0.016 | 0.953 | 0.948 | 0.012 | 0.717 | 0.790 | 0.065 | 0.787 | 0.832 | 0.069 |
| +LFNet [25] | 0.910 | 0.920 | 0.016 | 0.869 | 0.899 | 0.015 | 0.951 | 0.946 | 0.012 | 0.723 | 0.795 | 0.065 | 0.795 | 0.835 | 0.071 |

**Table 6.** The comparison of running speed of LIMS model fine-tuned with different optical flow models, the running speed is measured in frames per second(f/s), "+" indicates fine-tuning of LIMS using different optical flow models, "Ours" represents our optical flow model.

| Paltform | LIMS | +Ours | +CRAFT | +GMFlow | +PWCNet | +SpyNET | +LFNet |
|---|---|---|---|---|---|---|---|
| GTX1080Ti | 23 f/s | 24 f/s | 22f/s | 23f/s | 23 f/s | 23 f/s | 22 f/s |

### 5.2.2. Comparison with Current Mainstream VSOD Models

(1) Quantitative analysis of the performance changes brought about by fine-tuning the LIMS model using the optical flow models proposed and compare it with current mainstream video saliency object detection models, including QDFINet [48], PAC [49], LIMS [26], DCFNet [50], MQP [2], TENet [35], U2Net [51], PCSA [52], and LSTI [53]. The performance metrics on different datasets are compared in Table 7, where "+Ours" represents the metrics data after fine-tuning the LIMS baseline model with the proposed optical flow model in this paper, and the optimal data are shown in bold. After fine-tuning the LIMS baseline model with the proposed optical flow model in this paper, the performance metrics on the Davis [43] dataset are slightly improved. Due to the large and clear contours of salient objects in the Davis dataset, the object boxes obtained through the LIMS model can generate frame-level saliency maps with clear contours. The high-quality motion saliency maps generated by the new optical flow model in this paper have high consistency with saliency maps, in detail, which can help the LIMS model select high-quality saliency maps as keyframes to fine-tune image saliency detection models and generate the final saliency maps. However, the performance on the Segtrack-v2 [44] dataset slightly decreases; the Segtrack-v2 dataset is a dataset with diverse motion patterns and small significant objects. The saliency maps obtained through the LIMS model have higher consistency with low-quality motion saliency maps, which leads to the high-quality motion saliency maps obtained by the optical flow model in this paper not being able to help the LIMS model select high-quality keyframes to fine-tune the image saliency detection model. The performance metrics can be maintained stably across the Visal [45], DAVSOD [46], and VOS [47] datasets, as the frame-level saliency maps obtained through the LIMS model on these datasets exhibit similar quality. As a result, the high-quality motion saliency maps generated by the proposed optical flow model and the motion saliency maps generated by the traditional optical flow model, which select keyframes with similar quality, are unable to effectively improve the performance of the LIMS model, but it has improved the performance of the LIMS model to some extent.

**Table 7.** Comparison with current mainstream saliency detection models, "+Our" means fine-tuning LIMS using the optical flow model proposed in this paper, and the optimal data are shown in bold.

| Dataset | Davis [43] | | | Segtrack-v2 [44] | | | Visal [45] | | | DAVSOD [46] | | | VOS [47] | | |
|---|---|---|---|---|---|---|---|---|---|---|---|---|---|---|---|
| Metrics | maxF | S-M | MAE | maxF | S-M | MAE | maxF | S-M | MAE | maxF | S-M | MAE | maxF | S-M | MAE |
| QDFINet [48] | 0.912 | 0.918 | 0.018 | 0.834 | 0.883 | 0.015 | 0.952 | 0.946 | 0.012 | 0.705 | 0.773 | 0.069 | - | - | - |
| PAC [49] | 0.904 | 0.912 | 0.016 | 0.880 | 0.908 | 0.020 | 0.953 | 0.948 | 0.011 | 0.732 | 0.798 | 0.060 | 0.830 | 0.849 | 0.061 |
| DCFNet [50] | 0.900 | 0.914 | 0.016 | 0.839 | 0.883 | 0.015 | 0.953 | **0.952** | **0.010** | **0.791** | **0.846** | **0.060** | 0.660 | 0.741 | 0.074 |
| MQP [2] | 0.904 | 0.916 | 0.018 | 0.841 | 0.882 | 0.018 | 0.939 | 0.942 | 0.016 | 0.703 | 0.770 | 0.075 | 0.768 | 0.828 | 0.069 |
| TENet [35] | 0.881 | 0.905 | 0.017 | 0.810 | 0.868 | 0.025 | 0.949 | 0.949 | 0.012 | 0.697 | 0.779 | 0.070 | 0.781 | 0.845 | **0.052** |
| U2Net [51] | 0.839 | 0.876 | 0.027 | 0.775 | 0.843 | 0.042 | **0.958** | **0.952** | 0.011 | 0.620 | 0.728 | 0.103 | 0.748 | 0.815 | 0.076 |
| PCSA [52] | 0.880 | 0.902 | 0.022 | 0.810 | 0.865 | 0.025 | 0.940 | 0.946 | 0.017 | 0.655 | 0.741 | 0.086 | 0.747 | 0.827 | 0.065 |
| LSTI [53] | 0.850 | 0.876 | 0.034 | 0.858 | 0.870 | 0.025 | 0.905 | 0.916 | 0.033 | 0.585 | 0.695 | 0.106 | 0.649 | 0.695 | 0.115 |
| LIMS [26] | 0.911 | 0.922 | 0.016 | **0.899** | **0.921** | **0.013** | 0.953 | 0.947 | 0.011 | 0.725 | 0.792 | 0.064 | 0.822 | 0.844 | 0.060 |
| +Ours | **0.914** | **0.924** | **0.014** | 0.881 | 0.910 | **0.013** | 0.955 | 0.948 | 0.011 | 0.735 | 0.799 | **0.060** | **0.832** | **0.852** | 0.058 |

(2) Comparison of saliency maps generated by different VSOD models. In video scenes of different datasets, saliency maps generated by different model algorithms differ, as shown in the visual comparison results in Figure 7. The LIMS baseline model has already surpassed most of the current mainstream models, and after fine-tuning with the optical flow model proposed in this paper, the saliency map has been optimized in terms of details. For example, the video sequence "breakdance" in the Davis dataset, represented by the first and second rows in Figure 7, shows that models such as DCFNet [50], MQP [2], and TENet [35] are obviously disturbed by the background and generate saliency maps of objects in the background. Although the baseline model LIMS did not generate saliency maps of objects in the background, it generated foggy contours at the edge of the saliency map due to background interference. After fine-tuning LIMS with the optical flow model proposed in this paper, the generated saliency map has clear edge contours. In the "monkeydog" video sequence represented by the third row in Figure 7, fine-tuning the LIMS model with the optical flow model proposed in this paper makes the saliency map of the monkey more specific. In the "equestrian competition" video sequence represented by the fifth row in Figure 7, the salient pixel points inside the saliency map generated by the baseline model LIMS are missing, but after fine-tuning with the optical flow model proposed in this paper, the saliency map's interior becomes more substantial and reliable. Compared with other VSOD models, the LIMS baseline model solves the problem of background interference, but the saliency map's edges and interior exhibit different degrees of detail problems. After fine-tuning LIMS [26]with the optical flow model proposed in this paper, the LIMS model solves the problems of unclear saliency map contours and saliency map interior blur.

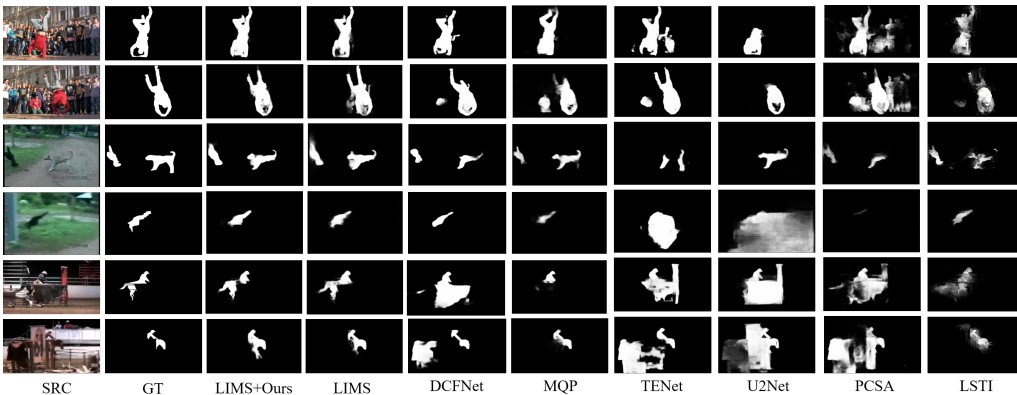

**Figure 7.** Comparison of saliency maps generated by different VSOD models, "LIMS+Our" refers to the saliency map generated by fine-tuning the LIMS model using the optical flow model proposed in this paper.

## 6. Conclusions

The optical flow model proposed in this paper can be utilized as a data augmentation technique to enhance the effectiveness of feature fusion, thereby improving the performance of video saliency detection models. Experimental results on five commonly used benchmark datasets show that the proposed optical flow calculation model can effectively improve the quality of motion information and that high-quality motion information can enhance the performance of the video saliency object detection model. The method presented in this paper sacrifices real-time processing to improve accuracy and is suitable for addressing problems with lower real-time requirements; in the future, we plan to improve the structure of the proposed optical flow model to minimize time and space consumption while obtaining high-quality optical flow maps.

**Author Contributions:** Conceptualization, H.W. and C.C.; methodology, H.W. and C.C.; software, H.W. and L.L.; validation, H.W., C.C. and L.L.; formal analysis, H.W.; investigation, C.P.; resources, H.W. and L.L.; data curation, H.W.; writing—original draft preparation, H.W.; writing—review and editing, L.L. and C.P.; visualization, L.L.; supervision, H.W.; project administration, H.W. and L.L.; funding acquisition, H.W. All authors have read and agreed to the published version of the manuscript.

**Funding:** This research is supported in part by the National Natural Science Foundation of China (No. 62172246), the Open Project Program of State Key Laboratory of Virtual Reality Technology and Systems (VRLAB2021A05) and the Youth Innovation and Technology Support Plan of Colleges and Universities in Shandong Province (2021KJ062).

**Data Availability Statement:** Not applicable.

**Conflicts of Interest:** The authors declare no conflict of interest.

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
