# Peer review of "Video Saliency Object Detection with Motion Quality Compensation"

_electronics, doi:10.3390/electronics12071618_

Round 1

Reviewer 1 Report

1. List out the importane of optical flow map

2. The motivation is to be improved further

3. Perform time and space complexity analysis and comapre the results with state of art

4. Explain in detail about the optical flow map towards the input whaich was taken. 

5. Mention a flow diagram or step by procedure that is used to conduct the proposed work.

6. Summarize your finding and compare with recent 2022&2023 articles 

Reviewer 2 Report

In this work the author proposed a model for optical flow computation. The topic is very interesting, and a lot of researchers proposed different network for solving this problem. Overall, the paper is well-written and organized, however, I have following comments the authors need to consider.

1.                The contribution of the work is not clear. The authors need to highlight what are the gaps left by the previous work and how they filled those gaps in the proposed one.

2.                Optical flow is usually prone to noises and illumination changes. How the proposed method deals that?

3.                Figure 2 should be modified as in the current form it is not showing the actual picture of the method.

4.                Experiment section is weak. Please compare the performance of proposed method with other similar methods.

5.                Discuss the failure cases and discuss the reason of failures

6.                The authors need to discuss the following related work based on an improved optical flow method.

a.       Analyzing crowd behavior in naturalistic conditions: Identifying sources and sinks and characterizing main flows. Neurocomputing 2016

7.                Discuss the application of proposed method in the revised version.

Reviewer 3 Report

In this paper, a new optical flow model is proposed to improve the quality of optical flow by expanding the range of optical influenza knowledge. Experimental results on the data set show that the proposed model can significantly improve the optical flow quality. This is an interesting research paper. There are some suggestions for revision.

1.   The motivation is not clear. Please specify the importance of the proposed solution.

2.   Please highlight the contributions/innovations of the proposed solution in introduction.

3.   Please discuss more recently published solutions, especially the solutions published in 2023 and 2022, such as "Small object detection method based on adaptive spatial parallel convolution and fast multi-scale fusion", Remote Sensing 14 (2), 420, 2022, and "Efficient Object Detection Based on Masking Semantic Segmentation Region for Lightweight Embedded Processors", Sensors, 22(22):8890, 2022.

4.   As for Figure 2 in the paper, there are some problems. Why does the model designed by the author only add color significance module to frame t, but not to other frames? Ask the authors to explain the advantages of this design.

5.   Similarly, there are still some doubts about Figure 2. Do the different colored lines in Figure 2 have different meanings? What is the operation of connecting frame t with other frames? Ask the author to explain it.

6.   For the parameters of formula 5 in the paper, there are some unclear points. It is mentioned in the paper that  in formula 5 is obtained by formula 4, but the specific process of obtaining parameter  is not seen in the paper. The author is asked to supplement it briefly.

7.   Please discuss how to obtain the suitable parameter values used in the proposed solution.

8.   There are some doubts about the ablation experiment of parameter n. Why the effect of high n value on some data sets is good, while that of low n value on some data sets is good? The author did not analyze the reasons in this paper, so the author is requested to make a supplement.

9.   As for the ablation experiment part of the paper, it seems that the authors have not completed the ablation experiment, so it is impossible to judge whether the module is really useful. Please think carefully to supplement.

10. For the method proposed in the paper, has the authors conducted experiments on the same type of data sets and achieved the same results as in the paper?

11. The experimental results are not convincing. Please compare the proposed solution with more recently published solutions.

Reviewer 4 Report

The paper presents a novel architecture for optical flow estimation with application to salient object detection. The architecture includes a module for color processing and a fusion module. I didn't find an ablation experiments showing the effect of removing this module, which surely has effect for slowly moving or static objects. Please discuss or add new results.

Other than that, the paper offers an honest piece of work with relatively exhaustive results.

Round 2

Reviewer 1 Report

The authors have satisfactorily addressed the questions.

Reviewer 2 Report

Thanks for considering and addressing my comments 

Reviewer 3 Report

All my concerns have been addressed. I recommend this paper for publication.